# Angular Distance-Guided Neighbor Selection for Graph-Based Approximate Nearest Neighbor Search

## Abstract

Graph-based approximate nearest neighbor search (ANNS) algorithms are widely used to identify the most similar vectors to a given query vector. Graph-based ANNS consists of two stages: constructing a graph and searching on the graph for a given query vector. While reducing the query response time is of great practical importance, less attention has been paid to improving the online search method than the offline graph construction method. This paper provides an extensive experimental analysis on the popular greedy search and other search optimization strategies. We also propose a novel angular distance-guided search method for graph-based ANNS (ADA-NNS) to improve search efficiency. The key innovation of ADA-NNS is introducing a low-cost neighbor selection mechanism based on approximate similarity score derived from angular distance estimation, which effectively filters out less relevant neighbors. We compare state-of-the-art search techniques, including FINGER, on six datasets using different similarity metrics. It provides a comprehensive perspective on their tradeoffs in terms of throughput, latency, and recall. Our evaluation shows that ADA-NNS achieves 34%-107% higher queries per second (QPS) than the greedy search at 95% recall@10 on HNSW, one of the most popular graph structures for ANNS.

## CCS Concepts

• **Information systems** → **Information retrieval query processing**; • **Theory of computation** → **Nearest neighbor algorithms**.

## Keywords

Approximate Nearest Neighbor Search, Similarity Search, Graph-based Approximate Nearest Neighbor Search

**ACM Reference Format:**
. 2018. Angular Distance-Guided Neighbor Selection for Graph-Based Approximate Nearest Neighbor Search. In *Proceedings of ACM Web Conference 2025 (WWW'25)*. ACM, New York, NY, USA, 10 pages. https://doi.org/XXXXXXX.XXXXXXX

## 1 Introduction

Nearest neighbor search (NNS), which returns database vectors with the smallest distance to the query vector, has been widely used in many application domains, such as database [19, 38], information retrieval [12, 25], machine learning [2, 10], and recommendation

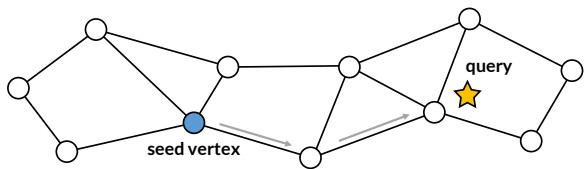

**Figure 1: Graph-based ANNS example**

systems [11, 28]. Exact solutions to this problem do not scale well to high-dimensional, large-scale data due to the phenomenon of the so-called *curse of dimensionality* [36, 41]. Approximate nearest neighbor search (ANNS) offers more practical solutions, retrieving $k$ nearest neighbors faster than the exact NNS algorithms at the cost of a slight accuracy loss. Significant practicality of ANNS has been drawn attention from production-level search engines, such as Microsoft search engine (e.g., Bing, Outlook, and SharePoint) [29] and Alibaba's Taobao e-commerce platform [16].

The typical ANNS algorihtms can be classified into four major types: tree-based [3, 6, 13, 17, 37], hash-based [1, 23, 26, 39], quantization-based [22, 24, 32], and graph-based ones [9, 15, 16, 21, 27, 31, 40]. The tree- and hash-based algorithms divide the vector space into multiple subspaces and index them using tree structures such as KD trees and hash tables. Quantization-based methods aim to reduce the complexity of distance computation through quantization. However, these methods often fail to retrieve accurate results due to the information loss or suboptimal partitioning of vector spaces. On the other hand, graph-based ANNS achieves both competitive query throughput and accuracy for large-scale datasets and various service scenarios [9, 21].

Graph-based ANNS consists of two essential phases—graph construction and search. To efficiently traverse large-scale vector datasets, it first constructs a graph that serves as the index for searching in the offline phase. Then, as shown in Figure 1, when the user queries a vector, starting from the seed vertices, the search algorithm gradually navigates through the connections to find vectors more relevant to the query vector. The search algorithm is typically implemented with *greedy* heuristics to find more relevant feature vectors step by step. The combination of graph construction and search algorithm plays an essential role in database query throughput and output quality (recall) compared to exact NNS.

Greedy search is typically deployed during the search, which evaluates similarity metrics of *all* neighbors at every selected vertex. While improving the query response time of graph-based ANNS is of great practical importance, less attention has been paid to improving this search algorithm than the graph construction method. Several proposals have addressed this problem by optimizing the similarity computation [8, 18, 31, 40], which is the major performance bottleneck. However, these approaches often fail to deliver a balanced solution in the trade-off between lower query response time and memory overhead.

To this end, we provide a detailed, quantitative analysis of a popular graph-based ANNS algorithm, HNSW [27], from various

perspectives. We observe that a large portion of computations are unnecessarily wasted, exposing opportunities to avoid costly similarity calculations. Building on these insights, we propose ADA-NNS, a novel angular distance-guided search mechanism integrated into a complete end-to-end graph ANNS system. Specifically, this paper makes the following contributions:

- We identify opportunities for significantly improving search efficiency with a minimal loss in precision by utilizing approximate similarity to filter out less relevant vertices on the search path. We find that only less than 20% of the neighbors at each vertex are pertinent to maintaining high accuracy with the various search algorithms and graphs.

- We propose ADA-NNS, a novel guided search method with advanced neighbor selection using approximate similarity score. We derive the lightweight proxy similarity from the angular distance between a neighbor vector and the query vector estimated using Sign Random Projection (SRP) [7]. We also design various optimization techniques to boost the performance of the guided search process effectively. Our evaluation demonstrates the robust performance of ADA-NNS across various datasets.

- We evaluate state-of-the-art search methods to identify trade-offs regarding throughput and recall. ADA-NNS achieves 34%-107% higher QPS than the greedy search on HNSW at 95% recall@10. These benefits come at a relatively small memory cost of 3.3%-11.7% of the original index, which includes both the graph and the vector dataset.

## 2 Background and Motivation

### 2.1 Approximate Nearest Neighbor Search

The objective of ANNS is to retrieve the most relevant $k$ vectors in the database to a given query vector. Most ANNS algorithms share a set of common similarity metrics correlating features of database vectors and the query, utilizing them to find better candidates. Consider a vector database $D$ with $d$-dimensional feature vectors $v$. For a given query $q \in \mathbb{R}^d$, ANNS finds candidate $k$ vectors in $D$ that maximize similarity metric $s$. Since the final $k$ vectors may not be accurate, recall@$k$ is used for measuring quality, which is defined as $\frac{|K \cap \hat{K}|}{|\hat{K}|}$, where $\hat{K}$ and $K$ are the set of the ground truth $k$ nearest neighbors and the set of the $k$ nearest neighbors returned by ANNS, respectively [14].

$$s_{L2}(q,v) = -||q - v||_2 = -\sum_{i=0}^{d-1}(q[i] - v[i])^2 \quad (1)$$

$$s_{IP}(q,v) = q \bullet v = \sum_{i=0}^{d-1}(q[i] \cdot v[i]) \quad (2)$$

**Similarity Metric.** L2 distance is one of the most commonly used similarity metrics (Equation 1). It is computed by summing the squared value of element-wise difference of $d$-dimensional vectors and then taking the square root of this sum. Since taking the squared root does not affect the relative ordering of L2 distance, it is often omitted. Additionally, negative values are used to indicate a stronger correlation for higher similarity scores. Another widely used metric is the inner product (Equation 2), computed as the dot product of

---

**Algorithm 1** Canonical Greedy Search Algorithm

**Input**: Graph $G$, query $q$, size of candidate list $efs$
**Output**: top-$k$ nearest neighbors in candidate list
1: $i = 0$, visited neighbors $V_n = \emptyset$, visited candidates $V_c = \emptyset$,
2: candidate $C$ = initial random seed vectors
3: **while** $i < efs$ **do**
4:     $V_c$.add($c_i \in C$)
5:     **for** $n_j$ in get_neighbor($\{c_i \in C \mid c_i \notin V_n \cup V_c\}$) **do**
6:         $V_c$.add($n_j$)
7:         $s$ = similarity($q, n_j$)       // Eq. 1 or 2
8:         $C$.add($n_j$)
9:         Sort $C$ in descending order of the similarity to $q$
10:         **if** $C$.size() > $efs$ **then**
11:             Drop the most irrelevant vector (low similarity) in $C$
12:         **end if**
13:     **end for**
14:     $i$ = index of the first unchecked vertex in $C$
15: **end while**
16: return top-$k$ IDs in $C$

---

vectors $q$ and $v$. This operation involves performing an element-wise multiplication of the two $d$-dimensional vectors, followed by a summation of the products.

### 2.2 Graph-based ANNS

Recent research in graph-based ANNS has focused on constructing high-quality graphs to improve search efficiency and precision. These approaches leverage various techniques to create graphs that ensure efficient connectivity with minimal edges, thus targeting high precision and low query response time.

**Canonical Greedy Search Process.** Algorithm 1 presents a canonical ANNS algorithm that outputs the top-$k$ closest vectors to a query vector. The search process begins by initializing the candidate list with either a given seed or random values (Line 1-2). Then, it enters the while loop, inspecting all the neighbors of the seed vectors $c_i$ in $C$. Candidates for which similarity scores have already been computed are marked as *checked* to prevent redundant calculations (Lines 3-4). Then, get_neighbor() fetches all neighbor vertices of the current candidate vectors, avoiding redundant similarity computation. Each iteration calculates the similarity between neighbor vectors and the query (Line 6), and the candidate list is updated and sorted accordingly (Line 9-10). This process iterates until all candidates in the candidate list have been fully explored.

### 2.3 Bottleneck Analysis of ANNS

**Breakdown of Query Response Time.** To analyze the detailed performance behaviors of ANNS, we decompose the search process described in Algorithm 1 into three stages. Initialization stage (Line 1), similarity computation (Line 7 in Algorithm 1), and the sorting stage that manipulates the candidate list, sorting, and candidate selection (Line 9-12 in Algorithm 1). Figure 2 presents the query response time on three representative datasets. Profiling results indicate that similarity computation takes a significant portion of query response time.

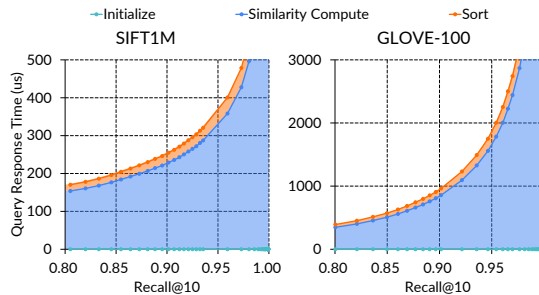

**Figure 2: Query response time breakdown of greedy search. The larger the area under the curve, the more severe the performance bottleneck it indicates.**

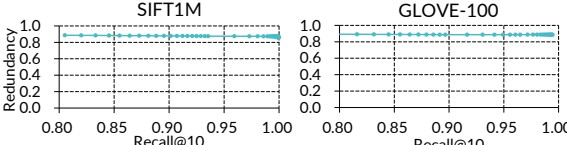

**Figure 3: Redundancy in the similarity computation of the greedy search. The higher the value, the less redundancy in the similarity computation.**

However, due to the high dimensionality of the vectors, similarity computation inherently requires significant computational resources and memory usage, rendering naive optimization approaches either ineffective or impractical. Moreover, we observe that similarity computations typically access each vector only once, resulting in poor data reuse and hence limited opportunities for cache-level optimization. Furthermore, it is challenging to improve memory latency with conventional techniques, such as prefetching, supported by commodity processors. In turn, the overall search process exhibits many irregular memory access patterns to high-dimensional data, which leads to substantial overhead and makes them impractical. Approximation of the similarity computation is one possible way to optimize, but naive approximation would suffer from significant accuracy loss.

**Redundancy in Similarity Computation.** A few recent proposals have attempted to avoid expensive similarity computation by filtering candidate neighbors [31, 40]. This selective neighbor calculation can be performed by extending `get_neighbor()` in Algorithm 1 (Line 4). Instead of returning all possible neighbors, some heuristics can choose only highly correlated ones for computation, functioning as `select_neighbors`. Unfortunately, however, research in this area remains relatively unexplored.

If a neighboring vector has a low similarity score, falling below the minimum candidate in the list, it will ultimately be discarded without ever entering the candidate list. This ensures that it does not influence the actual search path. Namely, such computation on highly uncorrelated neighbors just wastes resources. Moreover, this inefficiency is challenging to address during graph construction and requires better search strategies. Most graph construction methods are typically designed to provide high connectivity of vectors evenly, ensuring efficient search across a wide range of queries. As a result, if search algorithms do not adequately filter out target vertices, they should pay all costs of expensive computations. According to

our experiments across various datasets in Figure 3, more than 80% of the similarity computations are redundant during ANNS.

## 2.4 Related Work

Several strategies have been developed to improve the efficiency and throughput of the greedy search. The first approach emphasizes building an efficient graph while still employing the greedy search. The second approach is the guided search, which refers to algorithmic optimizations of canonical greedy search by filtering out similarity computations for less relevant neighbors. The third strategy focuses on using approximate similarity scores, which target reducing the computational cost of each similarity computation.

**Efficient Graph Construction.** The recent trend of optimizing graph-based ANNS has been reducing the number of similarity computations by graph construction methods. HNSW [27] organizes the graph into multiple levels: a coarse graph at the higher levels enables rapid navigation, while denser graphs at the lower levels facilitate precise searching. Thus, HNSW also consists of multi-level processes that continue searching in the lower-level graph once no more promising candidates are in the higher level. Other approaches such as NSG [16], Vamana [21], and NSSG [15] utilize single-level graphs, but their graph construction methods prune edges to build graphs that are sparse yet still maintain high connectivity. NSG and NSSG prune edges from a pre-built $k$-NN graph, which connects each element's $k$ nearest neighbors in the vector dataset. In NSG, by applying a similarity-based edge pruning method, the graph becomes sparse to improve search performance. NSSG takes into account the angle between vertices and similarity. Vamana [21] is the graph construction method used in DiskANN, which prunes edges from randomly initialized graphs. Vamana has relatively high edge selection flexibility connecting even vectors with lower similarity. Several studies [9, 21, 34] propose graph-based ANNS solutions optimized for slow memory (e.g., SSD) on memory-constrained environments (e.g., less than 64GB DRAM). These works incorporate optimizations designed to mitigate the high latency associated with slow memory. However, all of these approaches can be unified into a single canonical form of the search algorithm, as described in Algorithm 1, which remains suboptimal.

**Guided Search.** HCNNG [31] introduces the tree index to store the subspace information of neighbors. HCNNG computes the similarity only for neighbors in the query's direction using the subspace information. TOGG [40] uses either the KD-tree or K-Means clustering to store the distribution of neighbors. TOGG divides the search algorithm into two phases. The first phase aggressively filters out neighbor vectors to quickly move toward the query. Once the candidate is close enough, TOGG switches to the normal search phase, which investigates all neighbors. However, many less relevant neighbors still survive for downstream computations, which makes them perform a substantial amount of ineffectual exact similarity computation, yielding suboptimal throughput.

**Approximate Similarity Score.** This approach improves efficiency by adopting approximate similarity scores instead of exact similarity scores to reduce the number of operations required per similarity score. FINGER [8] improves query response time with minimal precision loss by estimating similarity through angular distance, using the current candidate as the center. ADSampling [18]

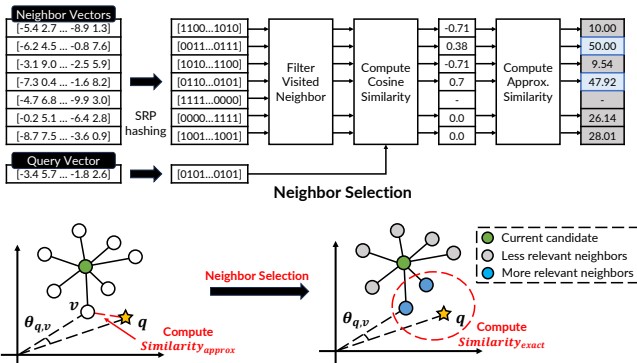

**Figure 4: Illustration of neighbor selection technique of ADA-NNS. It detects the neighbors toward the query.**

achieves similar precision with fewer computations by applying a random orthogonal transform to the dataset's vectors and adaptively sampling the dimensions needed for approximate similarity computation. However, FINGER incurs significant memory overhead to store the projected values for every edge on the graph. ADSampling is less suitable for CPU-based optimization techniques such as prefetching and SIMD due to its incremental dimension sampling scheme, which is hard to vectorize.

Our work focuses on in-memory vector database optimization, which targets fast query response time by holding all required data structures in memory. However, the proposed algorithm and implementation are not fundamentally restricted to this setting and are readily applicable to the scaled-up vector database.

## 3 Angular Distance-guided Search for Graph-based ANNS (ADA-NNS)

We propose ADA-NNS, featuring a novel guided search algorithm that dynamically filters out less relevant neighbors by employing a lightweight proxy based on an approximate angular distance. We first define angular distance and discuss its characteristics in Section 3.1. Section 3.2 then explains how ADA-NNS leverages this proxy metric to significantly reduce the number of operations without sacrificing accuracy. Finally, Section 3.3 consolidates the optimized search process with additional enhancements, delivering substantial performance improvements over state-of-the-art methods.

### 3.1 Estimating Similarity Scores with Approximate Angular Distance

In this section, we derive a lightweight approximation of similarity to filter out less relevant computations during the search process. Specifically, we will demonstrate that, when the approximate angular distance between the query and a neighbor exhibits minimal error. By using this lightweight metric, a subset of highly relevant neighbors can be identified at a low cost. To approximate two well-known similarity metrics, L2 distance and inner product discussed in Section 2, we first derive alternative forms.

**L2 distance.** Although L2 distance is described in Equation 1, we consider an alternative form:

$$s_{L2}(q, v) = -||q - v||_2$$
$$\stackrel{(a)}{=} 2 \cdot |q| \cdot |v| \cdot \cos \theta_{q,v} - |q|^2 - |v|^2$$
$$\stackrel{(b)}{\approx} 2 \cdot |q| \cdot |v| \cdot \cos \theta_{q,v} - |v|^2 \tag{3}$$

Here, (a) expresses the L2 distance in terms of the angle between vectors. This formulation can be further simplified because $|q|^2$ does not affect the relative ordering in the candidate list, and terms containing $v$ can be pre-computed offline. We emphasize that during the querying phase, the only term requiring computation is $\cos \theta_{q,v}$.
**Inner product.** Inner product in Equation 2 can be also written as:

$$s_{IP}(q, v) = q \bullet v = |q| \cdot |v| \cdot \cos \theta_{q,v} \tag{4}$$

Similar to L2 distance, $|v|$ can be pre-computed and reused, while $|q|$ is calculated only once for each query. Therefore, in Equation 4, the only important term is the $\cos \theta_{q,v}$.
**Approximate Angular Distance.** Both L2 and inner product (IP) share the critical term $\cos \theta_{q,v}$; thus, we can derive the approximated similarity score by estimating the $\theta_{q,v}$. Super-Bit Locality-Sensitive Hashing (SBLSH) [7] is well-suited for this objective. SBLSH is based on the Sign Random Projection, which maps vector $v \in \mathbb{R}^d$ to binary vector $b \in \{0, 1\}^m$, where $m \ll d$ capturing the difference in angles. We have carefully adapted SBLSH to estimate $\theta_{q}, v$ with parameters representing hash bit width $m$. To project vector $v$ into $b$, the hash function $h(v)$ performs the inner product between $m$ random orthonormal vectors $\in \mathbb{R}^d$. Once $b$ is generated for both query and vectors in the database, we can estimate the angular distance $\theta_{q,v}$ with the following equations:

$$\theta_{q,v} \approx \frac{\pi}{m} \times hamming(h(q), h(v))$$
$$hamming(h(q), h(v)) = popcount(h(q) \oplus h(v)) \tag{5}$$

Also, after retrieving $\theta_{q,v}$, we emphasize that the cosine values can be pre-computed and efficiently handled with a lookup table $T_{cos}$, which will be detailed in Section 3.3.

### 3.2 Neighbor Selection of ADA-NNS

**Neighbor Selection.** Using the estimated angular distance and the Equation 3 and 4, we compute an approximate similarity score. Algorithm 2 presents overall operation of `select_neighbor()` optimizing `get_neighbor()` in Algorithm 1. The algorithm outputs a selected set of $M$ neighbor vectors, which will undergo the exact calculation after return. After computing the approximate similarity scores (Line 8-9), the scores are added to the selection candidate set $S$ (Line 11). When the size of $S$ reaches its limit, the algorithm checks if any neighbor vectors exceed the minimum similarity score, $S[v_{min}]$, in the selected neighbor set $S$ (Line 15). Overall, we employ Sign Random Projection as a proxy metric for filtering neighbors, followed by the computation of exact similarity scores. Although it is theoretically possible to entirely eliminate exact score computations in Algorithm 1, our experiments reveal that doing so results in a significant decrease in accuracy. Therefore, we opt for a two-step candidate selection process instead.
**Preprocessing and Query Hashing.** Before starting ANNS, we perform the pre-computation mainly for $v$ and $q$ to reduce expensive operations, which is outlined in Section 3.1: the hash matrix $h$, the

---

**Algorithm 2** select_neighbor() of ADA-NNS

---

**Input**: Graph with $b_v \in \{0,1\}^m$, candidate $c$, # of neighbors $maxM$, query $b_q \in \{0,1\}^m$, unvisited set $UV$, cos table $T_{cos}$, param $\tau$
**Output**: $M$ neighbor vectors w/ highest approx. similarities

1:    // Starts with pre-computed $b_q, b_v, |q|, |v|, T_{cos}$
2:    List of selected neighbors and scores $S = \{\}$
3:    # of final selected neighbors $M = maxM \times \tau$
4:    **if** $UV$.size() $< M$ **then**
5:      return $S$                      // Early exit
6:    **end if**
7:    **for** all unvisited neighbors $v \in UV$ **do**
8:      $h\_dist = popcount(b_q \oplus b_v)$        // Eq. 5
9:      $s' = similarity_{approx}(|q|, |v|, T_{cos}, h\_dist)$    // Eq. 3 or 4
10:     **if** $S$.size() $< M$ **then**
11:       $S[v] = s'$
12:       **if** $S$.size() $= M$ **then**
13:         $b_{vmin} = $ Find $\text{argmin}_{b_v} S[b_v]$
14:       **end if**
15:      **else if** $S[b_{vmin}] < s'$ **then**     // Min-replace $b_{vmin}$
16:       Replace $b_{vmin}, S[b_{vmin}]$ with $b_v, s'$
17:       $b_{vmin} = $ Find $\text{argmin}_{b_v} S[b_v]$
18:      **end if**
19:    **end for**
20:    return $S$

---

norm of each vertex $|v|$, its square $|v|^2$, and the hashed dataset $H$ that consists of binary vectors $b_q$ and $b_v$ derived from $D$. We use the Gram-Schmidt Process [35] to generate $m$ orthonormal vectors $\in \mathbb{R}^d$ in the hash matrix and apply Equation 5. On the other hand, when a query arrives, ADA-NNS computes $|q|$ and the binary vector $b_q$. This stage is performed once per query, and $b_q$ is reused until it outputs the top-$k$ neighbors, thus still saving substantial computations.

## 3.3 Efficient Neighbor Selection of ADA-NNS

This section outlines the detailed optimization techniques of ADA-NNS to maximize the performance of angular distance-guided ANNS. Since the filtering operation in select_neighbor() introduces an additional step compared to the baseline get_neighbor(), analyzing the trade-off between the incurred overhead and performance gain is essential. We implemented the following optimizations to minimize such overheads, which will be quantitatively discussed in Section 4.

**Min-replacement.** The min-replacement algorithm maintains the selected neighbor list, simply replacing the lowest approximate similarity score $b$ with the new neighbor. As the output does not require sorting, this approach accurately identifies the top $M$ neighbors with significantly reduced computations. Specifically, the number of neighbors maintained in the selected list is limited to $M$ (Line 3 in Algorithm 2). If the selected list is not full, the neighbor and its approximate similarity score are added without computation (Line 10-11). Otherwise, the neighbor with minimum similarity should be identified (Line 12-18). Then, if the score of a new neighbor vector is higher than the minimum, it replaces the entry, and the minimum similarity is updated (Line 20-23). Min-replacement has a time

complexity of $O(M)$ for each replacement, offering a more efficient method than sorting, which has a complexity of $O(M \cdot log(M))$.

**Cosine Look-up and Early Exit.** To efficiently save the computation of $\cos \theta_{q,v}$, ADA-NNS introduces a cosine lookup table $T_{cos}$ exploiting a limited value range of hamming distance (0 to $m$), where each increment corresponds to an angular distance of $\frac{\pi}{m}$. Therefore, we can precompute a cosine lookup table $T_{cos}$ with $m + 1$ entries, updating the table values based on the Hamming distance index. This allows us to efficiently retrieve $\cos \theta_{q,v}$ without heavy computations. In addition, since the objective of select_neighbor() is to find $M$ vectors, if this condition is already satisfied, the function simply returns all the neighbor vectors (Line 4-6).

**Hyperparameter Tuning.** To find an optimized hash bitwidth $m$ that estimates $\theta_{q,v}$ with a sufficiently small error, we performed exploration measuring the standard deviation of errors from approximated angular distances. In each iteration, we increment $m$ by the multiple of SIMD width (e.g., 256-bit for AVX) to fully utilize data parallelism during distance computation. Our finding is that for $D$ with relatively small vector dimensions ($<= 300$), the standard deviation below 0.065 is enough to estimate the angular distance. For base sets with relatively high dimensions ($> 300$), a standard deviation below 0.035 is enough because of their high dimension. A sensitivity study on parameters is presented in Section 4.5.

**Neighbor Selection Threshold ($\tau$).** This parameter determines the final portion of the neighbors that are selected for exact similarity computation (Equation 1 and 2). The range of $\tau$ is from 0.0 to 1.0, where 1.0 means all neighbors are selected and 0.0 means no neighbors. Before computing the exact scores, the approximate similarity score is calculated using estimated angular distances. Then, the $\tau$ portion of the neighbors with the highest approximate similarity scores is selected for exact similarity computations. Setting high $\tau$ helps improve the recall by evaluating more neighbors. However, increasing the value excessively leads to a significant degradation in performance as it requires more similarity computations. We analyzed the detailed behavior of this parameter in Section 4.5.

## 3.4 Overhead Analysis of ADA-NNS

As mentioned in Section 3.2, ADA-NNS requires precomputation of a set of variables $|n|$, $|n|^2$, $h$, $H$, and $T_{cos}$. To store $|n|$ and $|n|^2$, we need $|D| \times 4$ bytes each, assuming each value is represented using a 32-bit floating-point. The size of the hash matrix $h$ occupies $m \times d \times 4$ bytes. The hashed set $H$ requires $\frac{m}{8} \times |D|$ bytes since each $m$ bit vector is packed into bytes. The cosine lookup table precomputes $(m + 1) \times 4$ bytes of data. Summing up all these, ADA-NNS requires $(8 + \frac{m}{8}) \times |D| + (m \times d + m + 1) \times 4$ bytes of memory. We also provide a detailed time complexity analysis on Appendix A.1

A million-scale dataset like SIFT1M (i.e., $|D| = 10^6$) with $m = 512$ results in an additional memory overhead of approximately 69MB. Considering that the total bytes of the less complex graph in SIFT1M is about 700MB, ADA-NNS presents a relatively small memory overhead of less than 10%. Note that this is significantly smaller than existing works, such as FINGER [8], which requires approximately 1300MB to store the projected values of all edges.

| Name | Dim. | No. of base | No. of query | Metric |
|------|------|-------------|--------------|--------|
| SIFT1M | 128 | 1, 000, 000 | 10, 000 | L2 |
| GIST1M | 960 | 1, 000, 000 | 1, 000 | L2 |
| CRAWL | 300 | 1, 989, 995 | 10, 000 | L2 |
| GLOVE-100 | 100 | 1, 183, 514 | 10, 000 | IP |
| NYTIMES | 256 | 290, 000 | 10, 000 | IP |
| DEEP100M | 96 | 100, 000, 000 | 10, 000 | L2 |

**Table 1: Datasets**

## 4 Evaluation

### 4.1 Experimental Setup

**Machine Configuration.** We conduct experiments on a machine with Intel i9-10920X CPU, 256GB DDR4 memory. All baselines are compile by g++ 9.4.0 with -O3 flag under Ubuntu 18.04 LTS. Microarchitectural optimizations including prefetching and SIMD are enabled.

**Datasets.** We use six different real-world datasets: SIFT1M [22], GIST1M [22], CRAWL [30], GloVe-100 [33], NYTIMES [4], and DEEP100M [5]. Table 1 summarizes the features of these datasets. SIFT1M and GIST1M are in the BIGANN dataset with images. SIFT1M dataset is trained with the SIFT descriptor, and GIST1M is trained with the GIST descriptor. CRAWL is the dataset trained using fastText. GloVe-100 dataset is the vector representation of words trained by GloVe algorithm. NYTIMES dataset is generated from the bag-of-words. To evaluate the scalability to larger datasets, we also perform experiments using DEEP100M [5] dataset. It is an image-based dataset, which is a subset of the DEEP1B dataset. DEEP1B is an image-based dataset trained with the DNN descriptor, which consists of one billion 96-dimensional floating-point vectors. The base set of DEEP100M consists of the first 100 million vectors in DEEP1B. It is the largest dataset that we can accommodate on our evaluation machine. Note that the size of DEEP100M vector dataset is about 37GB, which is much larger than other one-million scale datasets.

**Search Algorithms.** We compare the five algorithms whose descriptions are provided below:

- **Greedy Search (GS)** is the method that computes distances between all neighbors and queries. The greedy search can use various graphs. We use HNSW to compare with other baselines.
- **TOGG-KMC** [40] employs a two-phase guided search method. The first phase utilizes K-Means Clustering to create an additional data structure, which helps filter out less relevant neighbors to a query. The second phase then applies a greedy search algorithm. We use the authors' publicly available implementation[1], setting the number of neighbor clusters (*CN*) to 4.
- **ADSampling** [18] uses approximate similarity scores during search instead of exact ones. They preprocesses the vector dataset via random orthogonal transformation and adaptively samples the number of dimensions to compute the approximate similarity score. We run their open-sourced

code[2] to evaluate the performance using L2 distance only. Inner product similarity metric is not implemented.
- **FINGER** [8] is a practical search method that employs approximation to enhance efficiency. It initiates with exact similarity calculations for graph traversal during the first five iterations. Subsequently, it transitions to an approximated similarity metric, significantly reducing computational overhead. FINGER approximates similarity scores by estimating angles between neighboring vectors. In our implementation, we set *r*=64, which is the hyper-parameter for low-rank approximation of edge vectors.
- **ADA-NNS** is the novel guided search method proposed in this paper. Our method selects relevant neighbors based on the similarity score estimated from angular distance between vertex vector and query vector. Based on the observation in Section 2.3, we set $\tau = 0.2$, which denotes the ratio of relevant neighbors to compute exact similarity score. We set $m = 512$ for SIFT1M, CRAWL, GloVe-100, NYTIMES, and DEEP100M, while $m = 1024$ for GIST1M. Please refer to Section 3.2 for details about setting the hyper-parameter $m$.

**Metrics.** We evaluate the single thread performance of the search methods above in terms of queries-per-second (QPS) versus recall. QPS is inversely proportional to query response time, thus, for a certain recall, higher QPS is better. We measure the throughput at recall@10 and report the best throughput over 5 measurements.

### 4.2 Throughput Evaluation

Figure 5 illustrates the throughput improvements of ADA-NNS on HNSW across all datasets. ADA-NNS consistently outperforms other search methods in terms of throughput. Specifically, it achieves a 2.07× speedup over greedy search to reach 95% recall@10 on GIST1M, and a 1.71× speedup on CRAWL. FINGER generally performs better than greedy search but slightly worse than ADA-NNS. However, FINGER fails to execute on DEEP100M due to out-of-memory, which we discuss further in Section 4.6. TOGG-KMC shows mixed results, performing marginally better or sometimes worse than greedy search on some datasets. This poor performance can be attributed to the marginal reduction in similarity computations, as we explore in the following section. ADSampling consistently performs worse than greedy search across all datasets. Since their released code is not optimized for SIMD instructions, preventing it from leveraging the abundant data-level parallelism available when processing high-dimensional vectors.

### 4.3 Amount of Similarity Computations

In this section, we measure the amout of similarity computations to compare the algorithm efficiency of search algorithms. For ADA-NNS, the number of computations for approximate similarity computations are also counted. To achieve 95% recall@10 on GIST1M, ADA-NNS reduces the number of similarity computations to 41.6%, 33.5%, and 37.5% of those of greedy search method at 95% recall@10 on SIFT1M, GIST1M, and CRAWL. At 95% recall@10, we observe the ADSampling reduces the number of computation to 65.0%, 27.1%, and 54.9% of the greedy search for SIFT1M, GIST1M and CRAWL, respectively. Thus, the performance gains of ADA-NNS are attributed

---

[1]https://github.com/whenever5225/TOGG

[2]https://github.com/gaoj0017/ADSampling

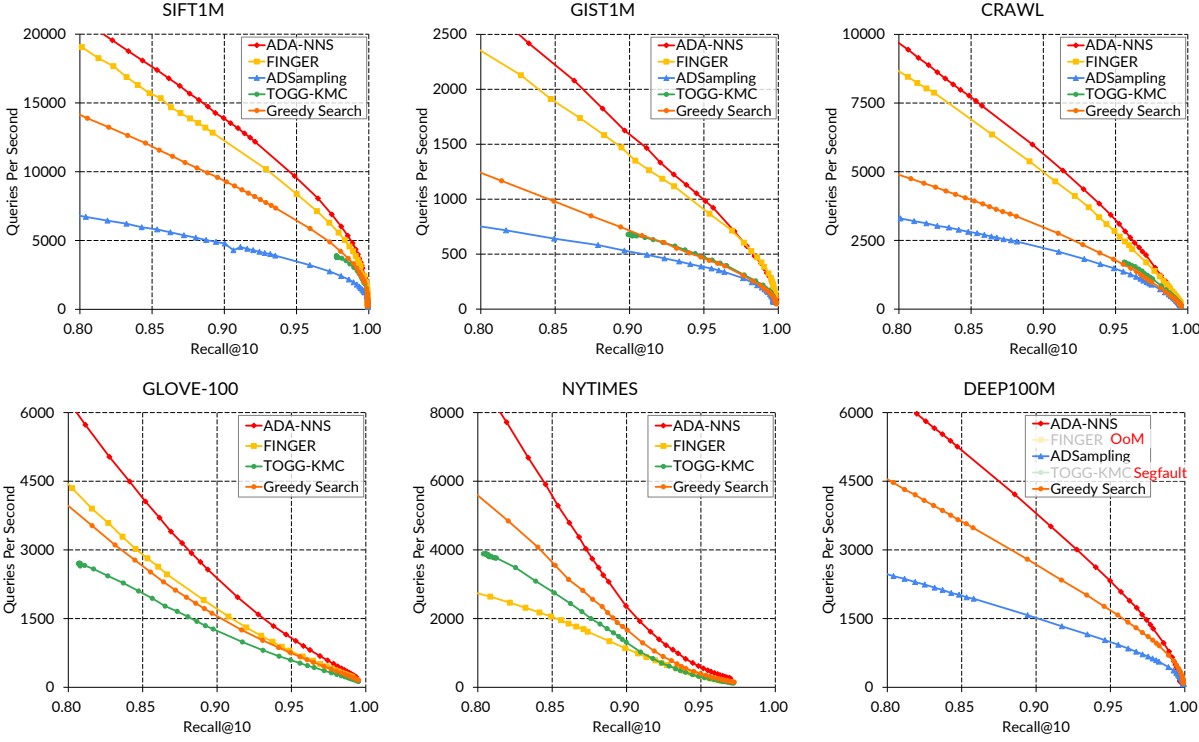

**Figure 5: Throughput versus Recall@10 plots for six datasets. SIFT1M, GIST1M, CRAWL, and DEEP100M use L2 distance as similarity metric. GLOVE-100 and NYTIMES use inner product as similarity metric.**

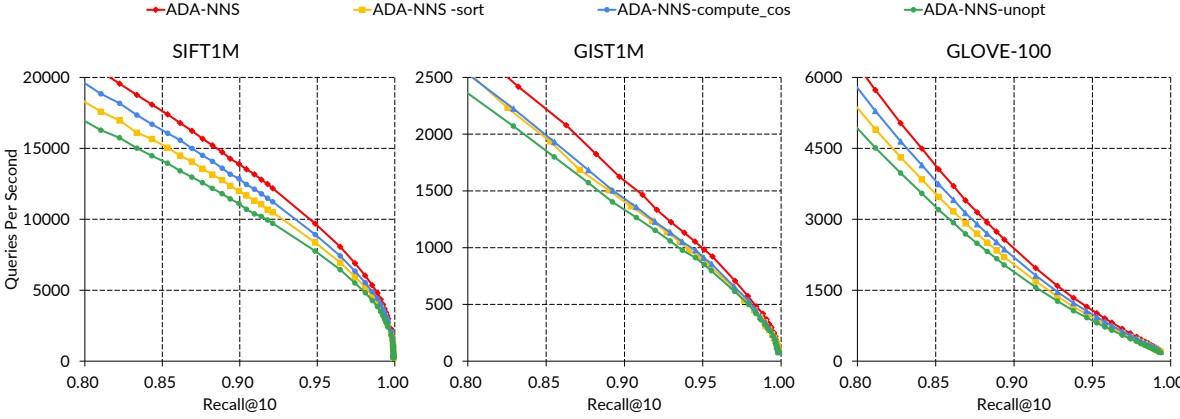

**Figure 6: Throughput versus Recall@10 results for ablation studies. ADA-NNS is the baseline with all optimizations. ADA-NNS-sort uses sort to select $\tau$ relevant neighbors. ADA-NNS-compute_cos uses cosine function instead of using cosine lookup table. ADA-NNS-unopt is ADA-NNS without optimizations.**

to the reduction in similarity computation. Despite ADSampling reducing computations more than our approach for GIST1M, it achieves lower throughput gains due to its algorithm design, which is not amenable to microarchitectural optimizations.

## 4.4 Ablation Study

In this section, we present an ablation study to analyze the effectiveness of two optimization techniques on the throughput gains

of ADA-NNS: min-replacement and cosine look-up with early exit. To demonstrate the impact of min-placement, we compare ADA-NNS with a variant (ADA-NNS-sort) that uses sorting to manage $\tau$ neighbors instead of the proposed min-replacement technique. To evaluate the technique of cosine look-up with early exit, we compare ADA-NNS with a variant (ADA-NNS-compute_cos) that uses the cosine function from the math library and disables the

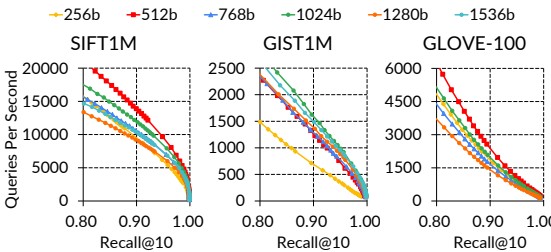

**Figure 7: QPS vs recall@10 plots of ADA-NNS on HNSW across three datasets with different hash bitwidth.**

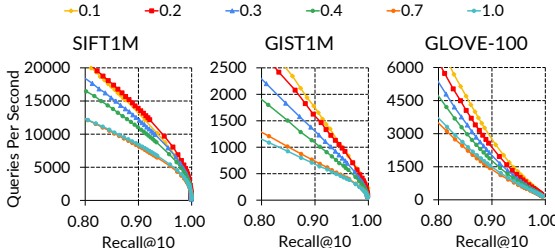

**Figure 8: QPS vs recall@10 plots of ADA-NNS on HNSW across three datasets with different $\tau$ values.**

early exit feature. We also include ADA-NNS-unopt, which is the approach without any optimization techniques, as a baseline.

Figure 6 presents the results of our ablation study. ADA-NNS outperforms ADA-NNS-sort by approximately 15% at 95% recall@10. This improvement can be attributed to the significant reduction in complexity achieved by using min-replacement instead of sorting, as described in Section 3.3. ADA-NNS shows an 8% improvement over ADA-NNS-compute_cos at 95% recall@10, demonstrating the effectiveness of the cosine look-up table and early exit strategy in reducing computational overhead. Overall, ADA-NNS achieves a 15.5%-25.0% better throughput at 95% recall@10 compared to ADA-NNS-unopt. These results confirm that both optimization techniques contribute significantly to improving the performance of ADA-NNS by reducing the overhead associated with our proposed neighbor selection method.

## 4.5 Sensitivity Study

**Hash Bitwidth ($m$).** In general, setting hash bitwidth ($m$) to a higher value would lead to higher precision in estimating the approximate similarity between two vectors. At the same time, it would increase i) the cost of hashing query, ii) the cost of computing approximate similarity, and iii) the required memory space to store the hash matrix $h$ and the hashed dataset $H$.

Figure 7 shows the QPS vs recall with varying $m$ across six different datasets. We set $\tau$ to 0.2. Setting $m$ to less than 512 (or 1024 for GIST1M) leads to lower throughput. Although a low $m$ may reduce the latency for hashing queries and computing hamming distances, it is not recommended. Because of the reduced precision of approximation of similarity score between the query and the neighbors of current candidate node, the search follows a suboptimal search path. The peak achievable QPS is almost saturated when $m$ is 1024 on GIST1M and 512 on the other datasets. Further increasing $m$ only

| Dataset | ADA-NNS | FINGER | TOGG-KMC |
|---|---|---|---|
| SIFT1M | 70MB | 1,289MB | 1MB |
| GIST1M | 134MB | 1,518MB | 1MB |
| CRAWL | 139MB | 2,034MB | 2MB |
| GLOVE-100 | 82MB | 1,526MB | 1MB |
| NYTIMES | 22MB | 217MB | 1MB |
| DEEP100M | 6,867MB | 148,011MB | - |

**Table 2: Memory overhead for compared baselines. TOGG-KMC fails on DEEP100M due to segmentation fault.**

marginally improves the precision of the approximated similarity score at best, while significantly increasing the computational cost.
**Neighbor Selection Threshold ($\tau$).** The appropriate setting of the neighbor selection threshold ($\tau$) is crucial for simultaneously achieving high recall and high QPS. Figure 8 shows the relationship between QPS and recall for various $\tau$ values. A $\tau$ of 0.2 indicates that only the top 20% of neighbors with the highest approximate similarity to the query are selected for exact similarity computation. Conversely, a $\tau$ of 1.0 prompts ADA-NNS to compute true similarity to the query for *all* neighbors of the current candidate vertex. Our results reveal that lower $\tau$ values lead to higher throughput but may significantly reduce recall if set too low, as relevant neighbors might be prematurely filtered out, whereas higher $\tau$ values enhance recall at the cost of reduced throughput.

## 4.6 Memory Overhead

Unlike greedy search, ADA-NNS employs approximate similarity scores for neighbor selection, necessitating additional data structures to estimate angular distances using Sign Random Projection. Table 2 illustrates the memory space required to store pre-computed values ($|v|$, $|v|^2$, $h$, and $H$) as described in Section 3.1. These pre-computed values occupy a relatively small portion (3.3%-11.7%) of the total index size, which includes both the graph and vector dataset. While FINGER achieves comparable throughput gains to ADA-NNS, it incurs significant memory overhead. For instance, FINGER requires some 288GB of memory to perform searches on the DEEP100M dataset, exceeding our experimental machine's capacity. Specifically, about 144GB is allocated to store auxiliary data structures, nearly equivalent to the space occupied by the original graph and vector dataset combined. In contrast, TOGG-KMC incurs negligible memory overhead, and ADSampling requires no additional memory. However, as demonstrated in Section 4.2, their performance gains are marginal or even inferior to greedy search.

## 5 Conclusion

This paper provides a quantitative analysis on the widely used greedy search method for graph-based ANNS. We propose ADA-NNS, an efficient search method on graph-based ANNS. We provide extensive experiment results across six million-scale datasets with different similarity metrics. Experimental results show that ADA-NNS not only achieves state-of-the-art throughput gain for various datasets with relatively small memory cost. Our approach does not modify the graph, which maintains good compatibility with existing graph construction methods.

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

# A Appendix

## A.1 Time Complexity Analysis

Time complexity of the greedy search on a graph is $O(c \times s \times l)$[27], where $c$ is the average number of neighbors on the proximity graph, $s$ is the search path length of greedy search, and $l$ is the number of graph layers, which is HNSW-specific parameter and set to 1 for other single layer graphs such as NSG [16], Vamana [21], and NSSG [15]. We can derive the time complexity of the ADA-NNS from that of greedy search from the following basis. First, we do not modify the graph, thus the parameters $c$ and $l$ does not change. Second, the filtering scheme in ADA-NNS targets on excluding exact similarity computations that would not be included in the candidate list, thus having no impact on the search path. Thus, we use time complexity of greedy search, denoted to $O_{GS}$, to derive the time complexity of ADA-NNS.

Neighbor Selection involves calculating the hamming distance, computing the approximate similarity, and replacing the entry with the minimum similarity in the selected list, followed by recalculating the new minimum. In the worst-case scenario, where all neighbors are unvisited and replacement occurs consistently in the selected list, the hamming distance and approximate similarity computation occurs $c$ times, while replacement occurs $0.8 \times c$ times. By utilizing bitwise operations on a 64-bit register (e.g., $uint64\_t$), hamming distance calculations consist of $\frac{m}{64}$ XOR operations, $\frac{m}{64}$ popcount operations, and $\frac{m}{64} - 1$ additions. When computing the approximate L2 similarity score (Equation 3), three multiplications and one subtraction are performed, followed by $0.2 \times c$ comparisons to determine the minimum similarity. In contrast, the exact similarity computation using L2 distance (Equation 1) involves $d$ subtractions, $d$ multiplications, and $d - 1$ additions.

Let $\bar{x}$, $\bar{p}$, $\bar{m}$, $\bar{a}$, $\bar{s}$, and $\bar{c}$ represents CPU cycles for XOR, popcount, multiplication, addition, subtraction, and comparison, respectively. The CPU cycles for Neighbor Selection is given by:

$$c \times (\frac{m}{64} \times (\bar{x} + \bar{p}) + (\frac{m}{64} - 1) \times \bar{a} + 3 \times \bar{m} + \bar{s} + 0.16 \times c) \quad (6)$$

Meanwhile, the CPU cycles of greedy search similarity computation is:

$$c \times (d \times (\bar{s} + \bar{m}) + (d - 1) \times \bar{a}) \quad (7)$$

According to Intel Optimization Reference Manual [20], CPU's cycles for serial execution of subtraction and multiplication is similar to that for serial execution of XOR and popcount, which means $\bar{x} + \bar{p} \simeq \bar{s} + \bar{m}$. By ignoring the minor impact of approximate similar term and comparator term, the complexity of Neighbor Selection is roughly $\frac{m}{64} \times$ that of the greedy search similarity computation, making its complexity approximately $\frac{m}{64 \times d} \times O_{GS}$. Furthermore, since the exact similarity computations in ADA-NNS are reduced by 80% through Neighbor Selection, the overall search complexity of ADA-NNS becomes as follows:

$$O_{ADA-NNS} \simeq (\frac{m}{64 \times d} + 0.2) \times O_{GS} \quad (8)$$