# OpenReview forum: "Angular Distance-Guided Neighbor Selection for Graph-Based Approximate Nearest Neighbor Search"
_ACM.org/TheWebConf/2025/Conference — WWW 2025 Poster_

### Official Review · Reviewer_wa2m · 2024-11-07

**Novelty:** 5
**Technical Quality:** 5

**Review:**

This paper introduces ADA-NNS, a new search algorithm tailored for graph-based Approximate Nearest Neighbour Search (ANNS) systems. ANNS algorithms goal is to efficiently find vectors similar to a query vector within large datasets. This has wide-ranging applications, including machine learning and information retrieval. The authors argue that existing methods for ANNS are suboptimal, particularly in their search algorithms, and introduce a novel approach using approximate angular distance as a filter, which effectively narrows the search space to include only the most relevant neighbors. The paper provides a robust experimental evaluation of the proposed algorithm, comparing it across multiple datasets and with various SOTA methods. Six real-world datasets of varying dimensions, sizes, and similarity metrics (including L2 distance and inner product) were used to benchmark the algorithm’s performance. The evaluation includes the DEEP100M dataset, which contains millions of vectors and offers a large-scale testbed to assess ADA-NNS's scalability across diverse applications, including image retrieval, text-based searches, and other data-intensive domains. The algorithm is compared to four search algorithms: Greedy Search (GS), TOGG-KMC, ADSampling, and FINGER using the Queries-Per-Second (QPS) metric at different recall levels (e.g., recall@10) and track the number of similarity calculations each algorithm requires. The authors provide further insights through an ablation study, examining the contributions of each optimization technique within ADA-NNS. Additionally, they explore the influence of key parameters, evaluate ADA-NNS’s memory overhead, and analyze the impact of parameters like hash bitwidth on algorithm efficiency. Overall, ADA-NNS achieves considerable throughput improvements, outperforming the baseline greedy search by 34%–107% at 95% recall@10 on the HNSW graph structure, all while maintaining minimal memory overhead.

Though ADA-NNS builds upon existing ANNS techniques, the combination of optimizations it incorporates - particularly the angular-distance-based filtering - are novel. The authors' detailed empirical evaluation highlights ADA-NNS’s effectiveness and positions it as a promising contribution to the field. While the experimental setup is described thoroughly, the addition of a GitHub repository with code and documentation on reproducing the experiments would enhance the paper’s usability and reproducibility.

### Strengths
(+) the paper introduces an angular distance-based filtering approach that efficiently excludes irrelevant neighbors, substantially reducing the number of exact similarity computations required. These gains are achieved without compromising search accuracy, maintaining comparable or improved performance relative to existing methods.
(+) the algorithm optimizes the search process independently of the underlying graph structure, ensuring compatibility with numerous existing graph construction techniques.
(+) the authors provide a thorough experimental analysis of ADA-NNS across six distinct datasets that represent varied domains and data characteristics, offering a holistic view of the algorithm’s strengths and limitations.

### Limitations
(-) the algorithm requires careful tuning of several hyperparameters for optimal performance. While the paper provides initial guidance, further research into automated or adaptive tuning strategies would likely enhance its practical usability.
(-) The evaluation is conducted on a specific hardware platform, leaving it unclear how ADA-NNS would perform across different hardware configurations.
(-) While the algorithm is benchmarked on HNSW, the evaluation could be expanded to include alternative graph structures like NSG, Vamana, and NSSG, offering a broader perspective on the algorithm’s generalizability.
(-) the algorithm is optimized for in-memory vector databases, but it would be valuable to examine its potential in disk-based or distributed environments, where data does not fully reside in memory.

**Questions:**

Questions
(?) How does the dimensionality of the data affect the optimal choice of hash bitwidth (m) in ADA-NNS?
(?) How scalable is the algorithm in environments where data cannot fit entirely in memory? Could the angular-distance-guided search effectively function in such scenarios, perhaps with minimal adaptation?
(?) Does the choice of graph construction method significantly impact ADA-NNS’s performance? Are certain graph structures more advantageous for ADA-NNS than others?
(?) How does the algorithm data updates? What are the computational and memory costs associated with modifying the necessary data structures?
(?) How does the algorithm perform in practical applications, such as recommendation systems, image search, and text search?
(?)  Could future work explore comparisons with other ANN methodologies, such as quantization-based approaches? This would offer insights into the algorithm’s versatility beyond graph-based systems.

**Reviewer Confidence:**

2: The reviewer is willing to defend the evaluation, but it is likely that the reviewer did not understand parts of the paper

**Scope:**

4: The work is relevant to the Web and to the track, and is of broad interest to the community

---

### Official Review · Reviewer_Y7iC · 2024-11-28

**Novelty:** 5
**Technical Quality:** 5

**Review:**

The ADA-NNS algorithm proposed in the paper is innovative in that it utilizes approximate angular distance for neighbor selection, but there is some content omitted from the detailed process description of the methodology. However, some processes in the methodology are somewhat omitted. Although the algorithm reduces the cost of similarity computations, the overall algorithm still involves multiple complex steps and components.

**Questions:**

The algorithm relies on a significant number of preprocessing steps during querying, including the construction of hash matrices and cosine lookup tables. For large datasets, do these preprocessing steps require substantial overhead? In scenarios where datasets need to be frequently updated, does the algorithm's real-time performance suffer excessively?

**Reviewer Confidence:**

2: The reviewer is willing to defend the evaluation, but it is likely that the reviewer did not understand parts of the paper

**Scope:**

3: The work is somewhat relevant to the Web and to the track, and is of narrow interest to a sub-community

---

### Official Review · Reviewer_EDEa · 2024-11-28

**Novelty:** 5
**Technical Quality:** 5

**Review:**

The paper focuses on the topic of graph-based approximate nearest neighbor search, which is a key ingredient of many vector databases. The authors introduce a new method called ADA-NNS that performs approximate similarity computations based on an angular distance metric, and they show that this approach achieves significant speedus with the same recall compared to greedy search on HSWN (one of the most commonly used implementation in modern frameworks).

The paper is well written, introducing the problem, relevant background and related work. The technical details are well explained and the experimentation seem sound, although statistical tests should be added to the experimental results.

Also, it would be nice to elaborate more on the bigger picture and connection to Web Search. Although it is true that approximate NN is a key ingredient of modern search engines, in the current shape the paper looks more suitable for a core data mining or database conference.

**Questions:**

see above

**Reviewer Confidence:**

3: The reviewer is confident but not certain that the evaluation is correct

**Scope:**

3: The work is somewhat relevant to the Web and to the track, and is of narrow interest to a sub-community

---

### Official Review · Reviewer_YDTy · 2024-11-30

**Novelty:** 5
**Technical Quality:** 5

**Review:**

This paper presents a novel angular distance-guided search method for graph-based ANNS (ADA-NNS) to improve search efficiency. A neighbor selection mechanism based on fast lookup operation is proposed to filter out less relevant neighbors. Experimental results shows that the proposed method boost existing grapg-based ANNS in terms of throughput, latency, and recall.

Pros:
1. This paper is well-written and easy to understand.
2. The proposed method can effectively reduce the redundancy during searching, which provide us a new way of thinking in accelerating the graph-based ANNs.

Cons:
1. Figure 3: "The higher the value, the less redundancy in the similarity computation". Is this conclusion reasonable?

**Questions:**

I am interested in the neighbor selection operation proposed in this paper, as it seems like a quantization and re-ranking process. In quantization-based ANNS methods, the distances between vectors can also be quickly approximated using lookup operations. Would it be reasonable to draw further comparisons between hashing-based and quantization-based methods?

**Reviewer Confidence:**

4: The reviewer is certain that the evaluation is correct and very familiar with the relevant literature

**Scope:**

4: The work is relevant to the Web and to the track, and is of broad interest to the community

---

### Official Review · Reviewer_JyXb · 2024-12-06

**Novelty:** 4
**Technical Quality:** 4

**Review:**

this paper presents a comprehensive analysis and innovation in the field of Approximate Nearest Neighbor Search (ANNS), focusing on graph-based methods.

The authors highlight that a significant portion of computations in traditional greedy search algorithms for ANNS are redundant and wasteful. They show that more than 80% of the similarity computations do not contribute to the results, indicating a substantial opportunity for optimization

he paper introduces ADA-NNS, a novel angular distance-guided search method. This approach uses approximate similarity scores derived from angular distance estimation to filter out less relevant neighbours. The method employs Sign Random Projection (SRP) to estimate these similarities, which is computationally lightweight and effective in reducing unnecessary similarity calculations.  By using approximate similarity scores, it reduces the computational cost associated with each similarity
 computation, leading to improved query response times without significant loss in precision


The authors conduct an extensive evaluation of ADA-NNS compared to state-of-the-art search techniques, including the greedy search on the HNSW graph structure. The results show that ADA-NNS achieves 34%-107% higher queries per second (QPS) than the greedy search at 95% recall@10, with a relatively small memory overhead of 3.3%-11.7% of the original index

timeliness
The paper's findings have significant practical implications for large-scale data applications, such as database query processing, information retrieval, machine learning, and recommendation systems. ANNS algorithms are crucial in these domains due to their ability to handle high-dimensional data efficiently

**Questions:**

My expertise on this topic is zero and I struggled to review this paper. Nevertheless, I have few queries/comments

section 2.3 - bottleneck analysis is presented without giving the details of the dataset, queries. Hence, i found it difficult to understand the material

the analysis of L2 overhead, finally has a component of cosine similarity (equation 3). This beg me to ask the question, why cant we use cosine similarity in teh first instance.

**Reviewer Confidence:**

1: The reviewer's evaluation is an educated guess

**Scope:**

2: The connection to the Web is incidental, e.g., use of Web data or API